# Comparative Analysis of SARS-CoV-2 Antibody Responses across Global and Lesser-Studied Vaccines

**DOI:** 10.3390/vaccines12030326

**Published:** 2024-03-19

**Authors:** José Victor Zambrana, Carlos Saenz, Hannah E. Maier, Mayling Brenes, Andrea Nuñez, Anita Matamoros, Mabel Hernández, Keyla Dumas, Cristhian Toledo, Leonardo Peralta, Aubree Gordon, Angel Balmaseda

**Affiliations:** 1Department of Epidemiology, School of Public Health, University of Michigan, Ann Arbor, MI 48109, USA; jzamb@umich.edu (J.V.Z.);; 2National Diagnostic and Reference Center, Ministry of Health, Managua 11165, Nicaragua; 3SILAS Managua, Ministry of Health, Managua 11165, Nicaragua; 4National Virology Department, Ministry of Health, Managua 11165, Nicaragua; 5Sustainable Sciences Institute, Managua 14006, Nicaragua; 6Department of Epidemiologic Surveillance, Ministry of Health, Managua 11165, Nicaragua; 7Department of Disease Prevention, Ministry of Health, Managua 11165, Nicaragua

**Keywords:** SARS-CoV-2, COVID-19, vaccines, comparative analysis, spike response

## Abstract

Few data are available on antibody response for some SARS-CoV-2 vaccines, and there is a lack of ability to compare vaccine responses in the same population. This cross-sectional study conducted in Nicaragua examines the SARS-CoV-2 antibody responses in individuals, previously exposed to high infection rates who have received various vaccines. The vaccines under comparison include well-known ones like Pfizer (BNT162b2) and AstraZeneca (ChAdOx1-S), alongside less-studied vaccines including Soberana (Soberana 02), Abdala (CIGB-66), and Sputnik V/Sputnik Light. Overall, 3195 individuals participated, with 2862 vaccinated and 333 unvaccinated. We found that 95% of the unvaccinated were seropositive, with much lower titers than the vaccinated. Among the vaccinated, we found that Soberana recipients mounted the highest anti-spike response (mean difference (MD) = 36,498.8 [20,312.2, 52,685.5]), followed by Abdala (MD = 25,889.9 [10,884.1, 40,895.7]), BNT162b2 (MD = 12,967.2 [7543.7, 18,390.8]) and Sputnik with AstraZeneca as the reference group, adjusting for age, sex, vaccine status, days after last dose, and self-reported COVID-19. In addition, we found that subjects with complete vaccination series had higher antibody magnitude than those with incomplete series. Overall, we found no evidence of waning in the antibody magnitude across vaccines. Our study supports the conclusion that populations with high infection rates still benefit substantially from vaccination.

## 1. Introduction

The vaccines against COVID-19 are a remarkable achievement in the fight against the SARS-CoV-2 pandemic. Effective vaccines for preventing hospitalization and severe disease were developed within less than a year after the identification of SARS-CoV-2 and have prevented millions of deaths [1,2].

In Nicaragua, the first case of COVID-19 was reported on 18 March 2020 [3]; by January 2022, thousands of cases had been reported, with three main pandemic waves [4]. Infection rates in that first pandemic wave, between April and July 2020, were high, with one study in Managua reporting 56.7% of the population was infected [5]. As seen globally, several variants circulated in 2021, including Alpha, Beta, Delta, and Gamma. A second large wave, predominantly of the Delta variant, peaked between August and September 2021 [6,7]. Omicron emerged in December 2021 [8].

More than 9 types of SARS-CoV-2 vaccines were developed globally, but only four—mRNA, viral vector, inactivated, and protein—became widely distributed [9]. Initially, in late 2020, most low- and middle-income countries, like Nicaragua [10], lacked access to these vaccines. This was particularly challenging for mRNA vaccines, which require ultra-low temperatures for storage and distribution, making them difficult to use in rural and lower-income areas.

In Nicaragua, several of the available vaccines were donated by the COVID-19 Vaccines Global Access Facility (COVAX) mechanism of the World Health Organization; Covishield was the first, in March of 2021 [11], followed by limited quantities of other vaccines, targeted initially at the elderly and high-risk populations. Additional vaccines were purchased by the Nicaraguan government or donated by individual countries, with a majority of the population receiving their first vaccine dose between June 2021 and November 2021, after most people had already experienced one or more infections [10,12,13,14].

Thus, three different types of vaccines, mRNA vaccines, viral vectored vaccines, and protein vaccines were obtained by Nicaragua and distributed to the population by age group. Vaccines distributed included Soberana (FINLAY-FR-2 and a final dose of FINLAY-FR-1A, RBD-tetanus toxoid [TT] conjugate protein vaccines), Abdala (CIGB-66, RBD protein vaccine), Covishield-AstraZeneca/Vaxzevria (ChAdOx1-S, viral vectored spike vaccine), Pfizer-BioNTech (BNT162b2, mRNA spike vaccine), Sputnik V (Gam-COVID-Vac, viral vectored spike vaccine), and Sputnik light (viral vectored spike vaccine). However, the immune response elicited by the various vaccine series has not been examined, and relatively few reports exist on the immune response to several of the vaccines administered to the population. Although the Cuban vaccines (Soberana and Abdala) were distributed in several countries, there are no published independent studies on these vaccines [15]. Of note, SARS-CoV-2 antibodies elicited through vaccination or infection protect against future severe infection and provide protection against disease when infected [16,17]. Higher levels of anti-spike antibodies are associated with increased protection from infection [17]. Here we examine the SARS-CoV-2 antibody levels in vaccinated and unvaccinated individuals in Managua, Nicaragua.

## 2. Materials and Methods

### 2.1. Study Area

The study was conducted in districts VI and VII in Managua, Nicaragua. A total population of 303,711 people lived in the study area, with 178,011 inhabitants in District VI (served by the Silvia Ferrufino and Roger Osorio Health Centers) and 125,700 inhabitants in District VII (served by the Villa Libertad Health Center).

The study area is of low and middle socioeconomic levels; neighborhoods included those near the coastal area of Lake Managua, semi-rural areas, and urban residential areas. All participants had access to a health unit. Nearly all (99%) of the population has access to water and sanitation. The recruitment of participants was carried out through home visits by the health team and community network.

### 2.2. Study Design and Data Collection

This cross-sectional study sampled Managua, Nicaragua’s population. Eligible participants were two years old or older, consented to participate, would not move from the area within six months, and, if vaccinated, had received at least one vaccine dose and showed their vaccination card. At enrollment, a blood sample was collected and a survey comprising demographic, health, vaccine data, and self-reported SARS-CoV-2 infection history was completed. Self-reported vaccination details were cross-verified with participants’ vaccination cards.

### 2.3. Sample

We used a non-probabilistic quota sampling method to select a sample of 3195 individuals from three health centers in Managua—Villa Libertad, Silvia Ferrufino, and Roger Osorio. The quotas for the health centers were 42% (*n* = 1330), 29% (*n* = 910), and 30% (*n* = 955), respectively. Each sample from each health center was divided into five age groups, with each age group being allotted 20% of the total sample. The age groups were 2–11, 12–17, 18–29, 30–59, and ≥60 years of age, and each age group was further subdivided into 90% vaccinees and 10% unvaccinated individuals. Sample distribution by health center population, age, and vaccines is shown in Appendix A.

### 2.4. Evaluated Vaccines and Age Groups

The vaccine series evaluated in this analysis are Soberana (2 doses of FINLAY-FR-2 [Soberana02] with a third dose of FINLAY-FR-1A [Soberana plus]), Abdala (3 doses of CIGB-66), AstraZeneca (2 doses of Covishield-AstraZeneca/Vaxzevria [ChAdOx1-S]), Pfizer-BioNTech (2 doses of BNT162b2), and Sputnik (2 doses of Sputnik V [Gam-COVID-Vac] or 1 dose of Sputnik light). Vaccine type availability varied with age. Complete vaccination series, including age, dosage and booster recommendations, are presented in Table 1.

Individuals were included regardless of the completeness of their vaccine series. Individuals were considered fully vaccinated 20 days after completing a vaccination series. Incomplete vaccination was defined as receiving at least one dose but not completing the series. Participants that received the same booster types after being fully vaccinated are considered as having a homotypic boost while those with an alternate vaccine have a heterotypic boost. We excluded 6 participants who did not complete their first vaccine series and received a different vaccine dose for our analyses, resulting in a final sample size of 3195.

### 2.5. Laboratory Methods

Anti-spike SARS-CoV-2 antibody levels were determined using the enzyme-linked immunosorbent assay (ELISA) method reported by Krammer et al. with some modifications (see Appendix A) [6,18]. All testing was performed at the Nicaraguan National Virology Laboratory.

### 2.6. Statistical Methods

Linear models investigated anti-spike titer levels. Adjusted models and crude ones were used. Analyses, pairwise and subgroup, were used to calculate mean differences in anti-spike magnitude, applying *t*-tests and linear regression. Population determinants of anti-spike antibody levels were investigated using age group, sex, vaccine type, vaccination status (complete, incomplete, boosted), dose number, and SARS-CoV-2 infection in a linear model. To avert statistical collinearity, dose number and vaccination status were separately analyzed. All tests were two-sided with an alpha level of 0.05.

The relationship between antibody levels and time since last vaccine dose was investigated in cross-sectional analyses. Population-level antibody dynamics used a multivariate linear model. Adjustments were made for age, sex, vaccine type, and SARS-CoV-2 infection. Interaction terms with time since last dose, vaccine, and vaccination status were considered. Anti-spike antibodies’ non-linear dynamics were modelled using GAM, with Gaussian distribution and thin-plate regression splines. Confidence intervals of 95% were calculated parametrically [19].

Subset analyses of antibody dynamics adjusted for age utilized linear modelling. The objective was to investigate changes in antibody levels over time per subgroup. Eligibility for both GAM and linear multivariate analyses required a minimum of 20 days post-vaccination. Confidence intervals and prevalence of 95% were calculated. Fisher tests examined prevalence differences per group. Unvaccinated groups had their antibody titer differences calculated by ANOVA.

All analyses were conducted in R version 4.2.1 (R Foundation for Statistical Computing, Vienna, Austria) and Epi Info 7.2 (C.D.C., Atlanta, GA, USA) [20,21].

## 3. Results

In January and February of 2022, 3195 individuals completed the serosurvey. Participants were distributed by the health center as planned (Appendix A). Overall, the five age groups were approximately equal; 64.7% (*n* = 2067) of participants were female, and 13% (*n* = 416) reported having had COVID-19 (Figure 1 and Figure 2, Appendix A). Most participants (89.6%, *n* = 2862) were vaccinated (Appendix A). Vaccination completeness was unequal by vaccine type. Vaccination was incomplete in 46% (*n* = 367) of AstraZeneca recipients, 72% (*n* = 575) of Abdala recipients, 51% (*n* = 138) of Pfizer recipients, 71% (*n* = 192) of Soberana recipients, and 19% (*n* = 133) of Sputnik recipients (Figure 2, Appendix A). In addition, only AstraZeneca and Sputnik recipients had received homotypic or heterotypic boosts.

### 3.1. Seroprevalence and Antibody Titers in Unvaccinated and N Antibodies in Vaccinated

Seroprevalence of anti-spike antibodies in the unvaccinated group was 95.2% (95% CI = [92.2%, 97.4%], Table 2). Seroprevalence was high in all age groups and both sexes, with no statistical differences between age groups (Table 2). Antibody titer magnitude was also similar by age group, sex, and study area with no statistical differences (Table 2).

Seroprevalence of N antibodies in a random selection of 100 vaccinated participants was 81.0% (95% CI = [71.7–87.9%]). Similarly, seroprevalence was high across age groups and sex, with no statistical differences between groups (Appendix A).

### 3.2. Antibody Titers

Anti-spike antibody titers were high in the general population, with a median titer of 56,692 (Appendix A). The vaccinated group had approximately five times the SARS-CoV-2 spike antibody titers of the unvaccinated group (median of 59,774 vs. 10,867, respectively, *p* < 0.001; Appendix A). After adjusting for age, vaccinated individuals still had much higher titers than the unvaccinated group (mean difference [MD] = 22,897.5, 95% CI = [18,779, 27,016], *p*-value < 0.001; Appendix A). In vaccinated individuals, children aged between 2 and 17 had higher anti-spike titers than older individuals in the univariate analysis. However, in the multivariable analysis of vaccinated participants, age was not statistically significant, suggesting that the age-specific differences were due to differences in the vaccines used rather than factors associated with age (Table 3).

Levels of antibodies among vaccinated individuals varied based on vaccine type. Soberana vaccinees had the highest anti-spike titers (MD = 36,498.8, 95% CI = [20,312.2, 52,685.5]), followed by Abdala vaccinees (MD = 25,889.9, 95% CI = [10,884.1, 40,895.7]) and Pfizer vaccinees (MD = 12,967.2, 95% CI = [7543.7, 18,390.8]). All three groups had higher anti-spike titers than AstraZeneca vaccinees (*p*-value < 0.0001 for each; Table 3) after adjusting for age, sex, vaccine status, days after last dose, and self-reported SARS-CoV-2 infection. Sputnik and AstraZeneca vaccinees had comparable titers (*p*-value = 0.959). We next compared the vaccine response across different age groups, and found higher anti-spike titers in 2–11-year-olds vaccinated with Soberana compared to those vaccinated with Abdala (*p* < 0.01) (Figure 3A). In 12–17-year-olds, antibody titers were comparable among Pfizer, Abdala, and AstraZeneca vaccines (*p* = 0.278). Among adults (18–29), Pfizer vaccinees had titers exceeding those vaccinated with AstraZeneca (*p* < 0.001) and Sputnik (*p* < 0.01), but no difference was observed between AstraZeneca and Sputnik vaccinees. These trends were repeated in the 30–59 age group. In populations over 59, AstraZeneca and Sputnik again showed similar responses (Figure 3A).

We then asked whether antibody titers varied by whether the vaccination series was complete (Table 3; Figure 3B,C). Individuals with a complete SARS-CoV-2 series had higher anti-spike titers than those who were incompletely vaccinated after adjusting for age, sex, vaccine status, days after the last dose, and self-reported SARS-CoV-2 infection (MD = 7181.5, 95% CI = [4122.8, 10,240.2], *p*-value < 0.001) (Table 3). Within vaccine types, higher anti-spike antibodies were observed in fully vaccinated recipients of Abdala, Soberana, and Pfizer compared to partially vaccinated ones (Figure 3B). However, for AstraZeneca and Sputnik, there was no significant difference in titers between fully and partially vaccinated individuals (Figure 3B,C).

Next, we examined whether the number of doses, regardless of vaccine type was associated with antibody titer levels. In univariate analysis, and after multivariable adjustment, vaccinees with two doses (*p*-value < 0.001) or three doses (*p*-value < 0.001) had significantly higher anti-S antibodies compared to those with one dose (Table 3; Appendix A). Interestingly, adjusting by dose number, Soberana vaccinees still had the highest mean magnitude (Appendix A). Within each vaccine series in a univariate analysis, increasing number of doses with increasing magnitude was only significant for Abdala (Appendix A). In addition, antibody titers did not differ by sex or by self-reported SARS-CoV-2 infection in either the univariate or the multivariate analysis (Table 3).

### 3.3. Antibody Dynamics

To examine waning and boosting over time, we analyzed antibody titers by the number of days since the most recent SARS-CoV-2 vaccine dose. In neither univariate nor multivariate analyses, antibody titers did not display significant waning over time since last dose (multivariate titer change of −0.5/day [−182.63/year], *p*-value = 0.222) (Table 3).

A stratified analysis, adjusting for age, showed no waning for most vaccine types and, in some instances, there was an increase (Figure 4). Sputnik vaccinees showed an increase of antibody levels over time that is statistically significant for both completely and incompletely vaccinated individuals with titer changes of 57.2/day [20,892.3/year] and 51.2/day [18,700.8/year], respectively (Appendix A, Figure 4). Homotypically boosted AstraZeneca and completely vaccinated Abdala vaccinees also had a substantial increase in the average antibody levels over time (Appendix A, Figure 4).

The multivariate analysis of antibody dynamics also demonstrated that, after adjusting for age, sex, type of vaccine, vaccination status, and self-reported COVID, anti-S antibodies neither increased nor waned significantly for up to a year post-vaccination for all vaccine types except for Sputnik (Appendix A). For Sputnik, a significant increase in titers over time was found (MD = 40.3/day [14,719.6/year], *p*-value = 0.0135). However, Sputnik baseline titer levels (at 20 days post-vaccination) were significantly lower compared to AstraZeneca baseline titer levels (MD = −8845.3, 95% CI = [−15,955.4, −1735.2], *p*-value = 0.0148) (Appendix A, Figure 4). In addition, we found no difference in antibody waning in completely vaccinated compared to incompletely vaccinated individuals.

## 4. Discussion

Here we found a high population level of immunity to SARS-CoV-2 in Managua, Nicaragua. While most unvaccinated individuals were already seropositive, vaccinated individuals exhibited even higher antibody titers. Our paper provides data on several vaccine types that are not used in the United States, the European Union, or the United Kingdom, and as such have many fewer or no population level studies available. Nicaragua experienced large SARS-CoV-2 waves before vaccinations; thus most vaccinated participants had hybrid immunity, a common situation for individuals vaccinated in low-income countries, further distinguishing our study. This research underscores that populations with high SARS-CoV-2 infection rates gain substantial benefits from vaccinations. The study did not assess vaccine efficacy, but antibodies identified through ELISA correlate with neutralizing antibodies [16,17,22,23,24,25], which is currently our correlate of protection for SARS-CoV-2.

Waning antibody response has been widely reported after vaccination against COVID-19 and natural infections [26,27,28,29,30,31]. However, vaccinated individuals with prior SARS-CoV-2 infection-induced immunity have shown a lower antibody decay rate [32]. We did not find significant waning for any of the studied vaccines or by vaccination status. Hybrid immunity and hyperendemic transmission of SARS-CoV-2 in Nicaragua may explain this, differing from high-income countries’ experience, in which confinement measures were applied and vaccines were available much sooner, and thus fewer people were infected prior to vaccination. We found an increase in antibody titers over time for Sputnik, which has been seen in other reports, especially after a second dose [33]. These high antibody levels in a population with hybrid immunity may suggest that populations with high levels of hybrid immunity may not need boosting as frequently to maintain the same antibody levels. Studies in vaccinated individuals with hybrid immunity have shown higher protection from disease and more durable response compared to vaccinated individuals only and have suggested a longer interval between infection and the booster for a higher immune response [34,35].

Our research offers valuable insights on several under-reported vaccines, comparing them to well-known ones like Pfizer and AstraZeneca. Notably, these are the first data on antibody responses to Cuban vaccines Soberana and Abdala outside of clinical trials [36,37,38]. In addition, to date, no studies on the long-term antibody kinetics exist on the Cuban vaccines, except for PastoCovac, an Iranian version of Soberana, which shows a stable IgG response up to 180 days [39]. Our study applied different vaccines to various age groups, with only children receiving Soberana and Abdala, and Pfizer compared to Abdala among 12–17-year-olds. Despite age not affecting the antibody levels in our study, prior research indicates a response decline with age for the AstraZeneca and Pfizer vaccines [40,41]. It is likely that Abdala and Soberana stimulate a strong immune response regardless of age. In particular, we hypothesize that the heightened immunogenicity of Soberana may be due to its use of tetanus toxoid and the near-universal tetanus vaccination coverage in Nicaragua [42].

In participants aged 18–59, Pfizer generated the highest antibody levels compared to Sputnik and AstraZeneca. These results match the literature; for instance, Adjobimey et al. found the combination with Moderna or BioNTech (Pfizer) mRNA vaccines induced significantly higher antibody levels than a double dose of AstraZeneca or Sputnik-V [43]. Also, Eyre et al. showed Oxford-AstraZeneca vaccine recipients had lower antibody levels after the first dose than Pfizer-BioNTech recipients, with or without previous infection [41].

Our study has many strengths, which include a large sample size and several vaccine types, including some with scarce evidence. It was conducted in a single population using the same assay for each vaccine. Also, including an unvaccinated group allowed for infection-induced seroprevalence data comparison. In addition, the inclusion of an unvaccinated group provides data on infection-induced seroprevalence and is an important comparison group. This study had some limitations. Assuming most vaccinated individuals have hybrid immunity based on SARS-CoV-2 spread and vaccination timing, we still lacked specific infection data for these individuals. While a subset of vaccinated individuals showed a high prevalence of N-antibodies, unclear timing between last infection and sample collection could affect interpretation of magnitude and decay rates in our cross-sectional study. Antibody dynamics can be best understood in longitudinal studies. Additionally, this study was designed on full and partial SARS-CoV-2 vaccine series, thus, our findings about boosters may be underpowered. And finally, the study utilized a non-randomized selection approach, potentially impacting the generalizability of the findings to broader populations. While our Spike ELISA correlates with neutralization, further research on neutralization titers against diverse variants would provide a more comprehensive picture of vaccine-specific immune responses.

During the SARS-CoV-2 pandemic, vaccines were not distributed equitably. Remarkably, many less resourced nations stepped up to the challenge and provided vaccines to low- and middle-income countries, as was seen in Nicaragua. However, fewer data were available on these vaccines at the time of rollout. While vaccine equity is certainly the ideal, the reality is that a similar situation may occur in future pandemics. In such cases, governments must prepare for evaluating the immunogenicity of all vaccines and ensure access to rigorously tested vaccines for a comparative analysis.

## 5. Conclusions

In summary, we found that all vaccines applied in Managua had a higher antibody response compared to SARS-CoV-2 unvaccinated, infected individuals. Unexpectedly, we found that Soberana recipients mounted the highest anti-spike response, followed by Abdala and Pfizer. Overall, we found no evidence of waning in the antibody magnitude over a year. Our study provides evidence that the primary SARS-CoV-2 vaccines administered in Nicaragua result in high antibody titers which suggests protection against SARS-CoV-2 in the Nicaraguan context, where most of the population have hybrid immunity and is fully vaccinated with approved and non-approved WHO vaccines.

## Figures and Tables

**Figure 1 vaccines-12-00326-f001:**
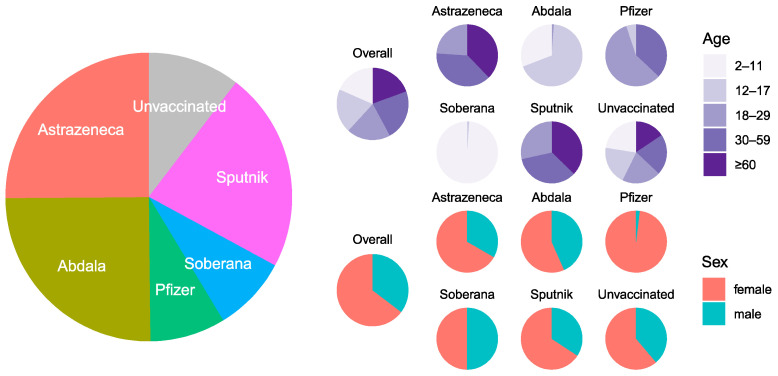
Distribution of vaccines by type, age, and sex. Participants characteristics by type of vaccine and unvaccinated. Relative distributions by type of vaccine and unvaccinated, including stratification by age groups and sex.

**Figure 2 vaccines-12-00326-f002:**
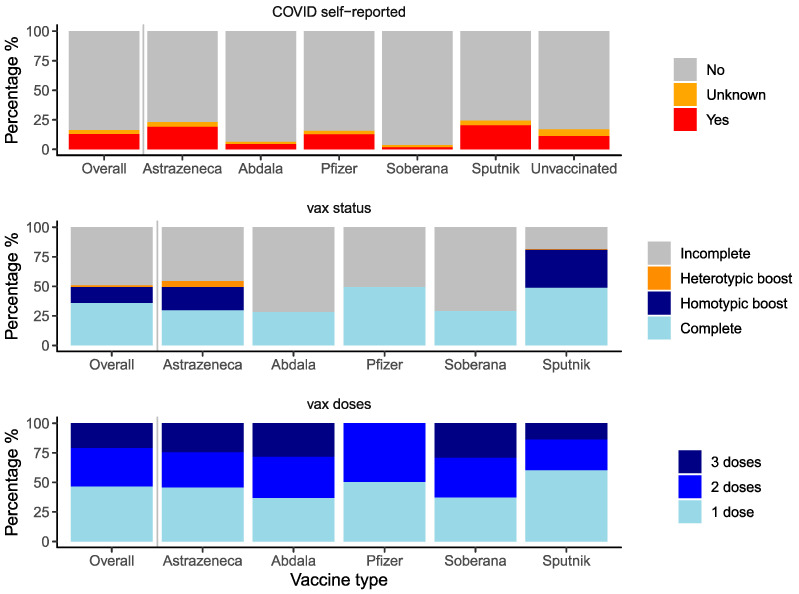
Distribution of self-reported COVID-19, vaccination status and number of doses. Distribution of self-reported COVID-19, vaccination status, and number of doses of the study population.

**Figure 3 vaccines-12-00326-f003:**
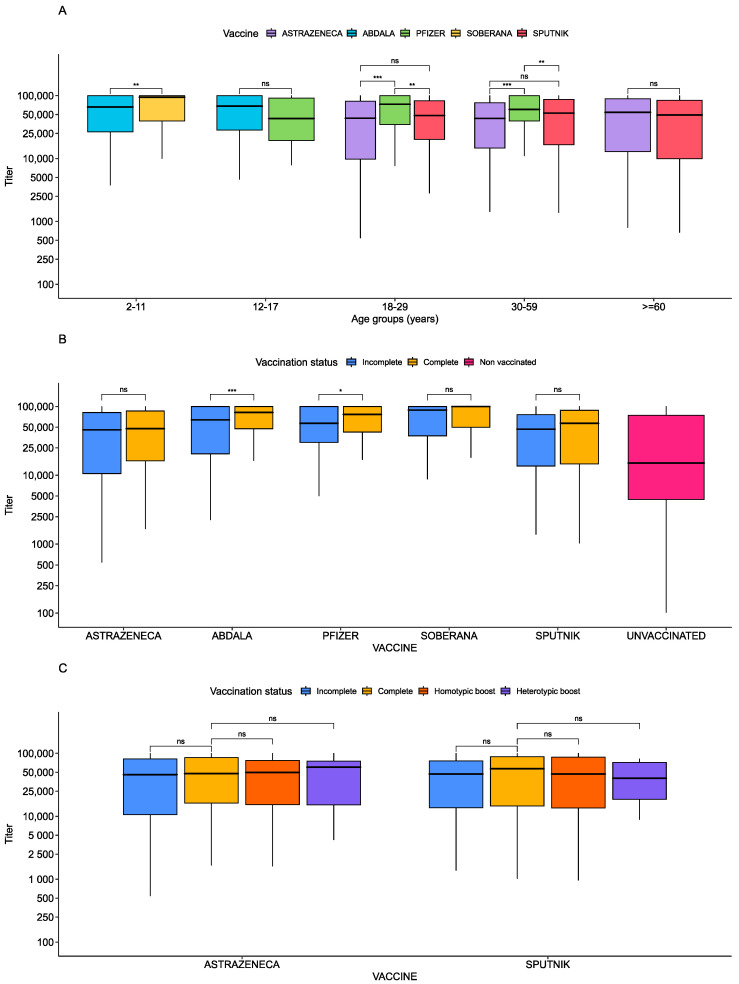
Anti-spike antibody titers per age and type of dose. Boxplots showing the anti-spike antibody magnitude by subgroups. (**A**) Antibody magnitude difference by type of vaccine and type of dose. (**B**) Antibody magnitude difference by type of vaccine and age, adjusted for intra-age differences. (**C**) Antibody magnitude by type of vaccine and type of dose and boosting. *T*-test *p*-values from pairwise comparisons are shown above boxplots. Significant *p*-values are shown with stars. ns is a *p*-value > 0.05, one star represents a *p*-value less than 0.05, two stars represent a *p*-value less than 0.01, and three stars represent a *p*-value less than 0.001.

**Figure 4 vaccines-12-00326-f004:**
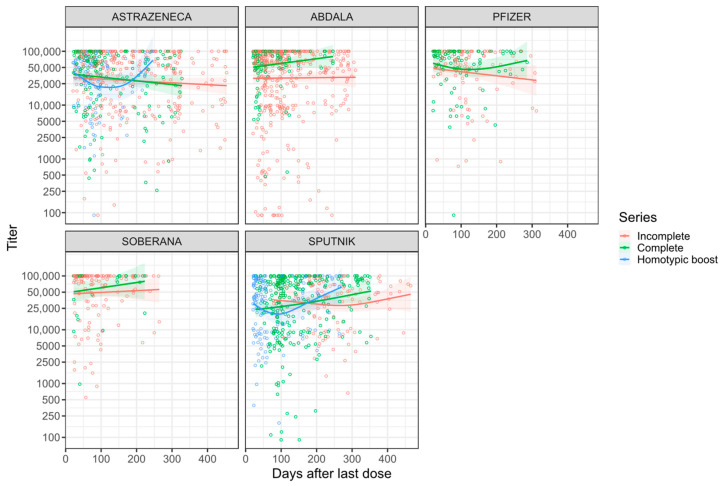
Anti-spike antibody kinetics by vaccine type. Non-linear relationship between anti-spike antibodies over time. Analyzed subgroups are the type of vaccine and dose. Colored lines are the GAM predictions, and grey bands are 95% confidence intervals.

**Table 1 vaccines-12-00326-t001:** Vaccine series evaluated in the study by dosage requirement, intervals, and age eligibility according to the Ministry of Health of Nicaragua.

	Dosage for Series Completion ^1^			Optional Boost	
Age Groups	Vaccine	Doses	Dosage Interval (Days)	Homotypic	Heterotypic
2–11 y/o	Soberana (FINLAY-FR-2) ^2^	3	28	-	-
2–17 y/o	Abdala (CIGB-66) ^3^	3	28	-	-
≥12 y/o	Pfizer-BioNTech (BNT162b2)	2	21	-	-
≥18 y/o	Sputnik V (Gam-COVID-Vac)	2	21	Sputnik Light	Covishield or Vaxzevria
≥18 y/o	Sputnik light	1	-	Sputnik Light	Covishield or Vaxzevria
≥18	Covishield-AstraZeneca (ChAdOx1-S)/Vaxzevria (ChAdOx1-S) ^4^	2	120	Covishield or Vaxzevria	Sputnik Light

^1^ Fewer doses than required for series completion were considered incomplete dosages. ^2^ The Soberana series consists of two doses of FINLAY-FR-2, followed by a third dose of FINLAY-FR-1A (also called Soberana Plus). All doses are administered at a 28-day interval, and completion of the series requires all three doses. ^3^ The Abdala series consists of three doses of CIGB-66. All doses are administered at a 28-day interval, and completion of the series requires all three doses. ^4^ The AstraZeneca series consists of two doses of either Covishield-AstraZeneca (ChAdOx1-S) or (ChAdOx1-S)/Vaxzevria (ChAdOx1-S) in any order. Both vaccines are the same but sold under two different brands.

**Table 2 vaccines-12-00326-t002:** Seroprevalence and antibody titer distribution in unvaccinated participants by age, sex, and health center population.

Variable	Categories	N	Seroprevalence (%)	Fisher Test *	Median Titer (IQR)	Anova *
Overall	-	333	95.2 (92.2–97.4)	-	10,867 (3330–70,641)	-
Age group	2–11 y/o	75	90.7 (81.1–95.8)	0.2562	9054 (1286–57,433)	0.42
	12–17 y/o	67	97.0 (88.7–99.5)		8284 (2837–78,859)	
	18–29 y/o	68	98.5 (91.0–99.9)		11,912 (4982–75,010)	
	30–59 y/o	71	95.8 (87.3–98.9)		14,352 (4039–68,012)	
	>60 y/o	52	94.2 (83.1–98.5)		22,280 (6250–89,087)	
Sex	Female	204	96.1 (92.1–98.2)	0.46884	9821 (3903–71,550)	0.883
	Male	129	93.8 (87.8–97.1)		11,627 (2723–67,046)	
Health Center	Roger Osorio	108	96.3 (90.2–98.8)	0.08052	10,662 (4448–77,328)	0.088
	Silvia Ferrufino	85	90.6 (81.8–95.6)		6233 (1268–51,066)	
	Villa Libertad	140	97.1 (92.4–99.1)		19,448 (5040–76,761)	

* Bivariate analyses.

**Table 3 vaccines-12-00326-t003:** Risk factor analysis of anti-spike antibody magnitude in vaccinated participants.

				Univariate Analysis		Multivariate Analysis ^1^	
Covariate	Category	n	Median Titer	Model Estimate	*p*-Value	Model Estimate	*p*-Value
Age groups	2–11 y/o	515	80,940	14,300.9 [10,013.3, 18,588.6]	<0.001	−15,335.4 [−31,112.9, 442.0]	0.057
	12–17 y/o	560	67,498	10,670.0 [6473.8, 14,866.2]	<0.001	−13,642.1 [−28,749, 1464.9]	0.077
	18–29 y/o	563	55,200	2851.8 [−1338.7, 7042.4]	0.18	−2323.6 [−6821.7, 2174.4]	0.311
	30–59 y/o	654	50,514	89.9 [−3951.3, 4131.1]	0.97	−2885.9 [−7053.5, 1281.6]	0.175
	≥60 y/o	570	51,712	Ref.	-	Ref.	-
Sex	Male	999	58,195	Ref.	-	Ref.	-
	Female	1863	60,555	1638.6 [−1160.2, 4437.4]	0.25	2783.1 [−88.1, 5654.3]	0.058
Type of vaccine	AstraZeneca	802	47,914	Ref.	-	Ref.	
	Abdala	801	67,501	12,033.5 [8534.9, 15,532.1]	<0.001	25,889.9 [10,884.1, 40,895.7]	<0.001
	Pfizer	272	67,368	12,928.7 [8014.4, 17,843.0]	<0.001	12,967.2 [7543.7, 18,390.8]	<0.001
	Soberana	270	94,062	21,211.7 [16,283.8, 26,139.6]	<0.001	36,498.8 [20,312.2, 52,685.5]	<0.001
	Sputnik	717	49,048	1426.3 [−2173.3, 5026.0]	0.44	100.5 [−3714.4, 3915.3]	0.959
Vaccination status	Incomplete	1405	59,933	Ref.	-	Ref.	
	Complete	1027	66,362	3144.5 [223.7, 6065.2]	0.0349	7181.5 [4122.8, 10,240.2]	<0.001
	Homotypic boost	387	47,700	−6585.3 [−10,669.6, −2501]	0.0016	2105.7 [−2537.3, 6748.6]	0.374
	Heterotypic boost	43	60,413	−6108.3 [−17,122.5, 4905.8]	0.2771	956.8 [−10,600.0, 12,513.5]	0.871
Days after last dose	Titer change by day	-	-	−0.6 [−1.5, 0.3]	0.19	−0.5 [−1.4, 0.3]	0.222
Number of doses	1	1334	54,096	Ref.	-	-	-
	2	932	65,250	7160.9 [4127.3, 10,194.6]	<0.001	-	-
	3	596	64,768	7062.2 [3561.2, 10,563.3]	<0.001	-	-
Self-reported SARS-CoV-2 infection	Yes	378	60,392	Ref.	-	Ref.	-
	Unsure	90	53,623	−5592.7 [−13,964.7, 2779.4]	0.19	−5602.9 [−13,984.4, 2778.7]	0.19
	No	2393	59,861	614.9 [−3335.9, 4565.6]	0.76	−2176.4 [−6512.7, 2159.8]	0.325

^1^ Linear model adjusted by age group, sex, vaccine type, and days after last dose.

## Data Availability

The study’s data is not public due to confidentiality concerns. Data is available upon request from the corresponding author [A.G.].

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
