# Peer review of "Comparative Analysis of SARS-CoV-2 Antibody Responses across Global and Lesser-Studied Vaccines"

_vaccines, 2024, doi:10.3390/vaccines12030326_

Round 1

Reviewer 1 Report

Comments and Suggestions for Authors

Ref. VACINES_2850506_reviewer

 Article 1 

“ Comparative Analysis of SARS-CoV-2 Antibody Responses across Global and Lesser-Studied Vaccines”

Abstract

“Question 1: line 24 “poorer, and vulnerable populations” – Does this information change the context of the study?

Introduction

Question 2: line 51 “Omicron emerged in December 2021.” Suggestion: include the sentence in the previous paragraph.

Question 3: line 52 – 56 “Five SARS-CoV-2 vaccines are developed globally, with four types widely distributed: mRNA, viral vectored, inactivated, and protein vaccines. Initially in late 2020, most 53 low and middle-income countries, like Nicaragua [9], lacked access to these vaccines. mRNA vaccines, needing ultra-low temperatures, posed a challenge especially in rural, lower-income areas.” Suggestion: rewrite this paragraph, so that the information is more direct and in the correct tense. 

Question 4: line 65 – 79 – Suggestion: Place in a separate paragraph; this paragraph was too long.

Culture of strains 

Question 5: line 86 – 90 (2.1. Study area): The objective of the study was to evaluate the difference in vaccination and immune response in the low and middle income population? Was it to assess whether the economic condition could be an influencing factor for the immune response? If this is the context, leave this better described in the introduction, in the methodology and highlight this in the title.

2.3. Sample (line – 99 a 105)

Question 6: Line 99 – 101 (2.3. Sample) – “ We used a non-probabilistic quota sampling method to select a sample of 3,200 individuals from three health centers in Managua - Villa Libertad, Silvia Ferrufino, and Roger Osorio. The quota for each health center was 41%, 29%, and 30%, respectively”.  Suggestion;  “Arrange the N = 3,200, according to the number used in the tables, results and statistical analysis.”

2.4. Vacinas avaliadas e faixas etárias (line 107 – 111)

Question 7: line 107 - 111 (2.4. Evaluated vaccines, and age groups):  

Suggestion: The Soberana plus vaccine (FINLAY-FR-1A) is not mentioned in table 1; If it was not used in the analyses, I suggest removing it from the text. Standardize the identification of vaccines in all tables (table 1, table s1; table S4; table S5), always using the same identification, as it is confusing as it is.

3. Results

Question 8: line 223 - 227 “ Figure 1. Distribution of vaccines by type, age, and sex. Participants characteristics by type of vaccine and unvaccinated. Relative distributions by type of vaccine and unvaccinated, including stratification by age groups and sex.”  Suggestion: I suggest removing this figure from the article, as it does not contribute anything to the data and is not mentioned anywhere;

Question 9:  line 244 – 249 (Table 1; table 2; table 3)

Suggestion: “Describe in detail the title of the tables; It was very poor the way it was placed.” Fill all columns with values, even if no number appears (insert a hyphen) – table 1 and 2.

4. Discussion

Question 12: line 265 “ we did not find significant waning.”  

Suggestion: Discuss this data found further, as the research group has data for this.

Question 11: line 272 – 275 “These high antibody levels in a population with hybrid immunity may suggest that populations with high levels of hybrid immunity may not benefit substantially from boosters of the original vaccine strain or may not need boosting as frequently to maintain the same antibody levels.” 

Suggestion: This statement can be supported by studies that evaluated this; use survey data for this statement, contextualizing the research group’s statement.

Supplementary: 

Question 12: Table S1. Participant characteristics. 

Suggestion: “Describe in detail the title of the tables; it was very poor the way it was placed.”

Reviewer 2 Report

Comments and Suggestions for Authors

This manuscript presents a cross-sectional study of antibody responses to SARS-CoV-2 spike antigen in >3000 individuals in Nicaragua who received different COVID-19 vaccines or were unvaccinated. The strength of the study comes from the large study cohort and the many different vaccines that were used in this population, including many vaccines that were deployed in many low- and middle-income countries but have not been much studied. Comparation of vaccines based on various platforms (e.g. recombinant protein subunit, protein conjugate, adenovirus vectored, mRNA) demonstrated their ability to elicit varying levels of spike-specific antibodies. However, their relative magnitudes and waning rates are not so straight forward to define. The authors’ conclusions in the Abstract and the rest of the manuscript should be qualified as there are confounding factors that should be taken into consideration.  The specific comments are listed below.

Major comments:

-              Only spike-specific antibodies were measured. It would be more informative if antibodies against N (nucleocapsid) were also detected to assess the prevalence of virus exposure/infection among the vaccinated groups. If the authors assume the 95% seroprevalence seen the unvaccinated to be applicable to the vaccinated groups, this assumption should be clearly stated and incorporated in the data analyses as the so-called hybrid immunity is associated with distinct immune responses.

-              “Complete” vaccination series vary from 1 to 3 doses depending on the vaccine types. Comparison of antibody magnitude (e.g. Figure 3) did not take this variable into consideration. For example, the higher levels mounted by Soberana02 and Abdala vaccines may be related to the 3 doses required to complete these vaccine types. The age groups are also very different; 2-17 for these two vaccines.

-     The time of sample collection relative to the last dose for each of the groups is not understood; the data need to go into Table 3, be considered in the data analysis, and be specified in most of the summary statements in the Abstract. These data seem to be included in Table 1S, but each mean has a tremendous SD (e.g. 94.4 +/-1549.8). Would it be useful to also show the range? 

-              The authors found no evidence of waning in the antibody magnitude across the vaccine types. In addition to Figure 4, is it possible to compare the antibody titers at several time periods (e.g. <100, <200, <300 days)? Nonetheless, the kinetics of antibody response is best measured in a longitudinal study. Exposure to the virus to the vaccine recipients between vaccination and sample collection would also boost antibody responses and prevent the ability to detect any decline during the study period.

-              Lines 112-118: Table 1 should be explained more clearly especially the doses required for complete vaccination for each vaccine type and how these differ from the additional boosts given to some of the groups. This information should also be made clear throughout the manuscript. Dose 3 for one vaccine may be part of the complete series or a booster for another vaccine; this may have a significant impact on antibody magnitude as mentioned above.

Minor comments:

Line 46: “since then” should be specified by a specific time (month/year).

Line 52: There are more than five SARS-CoV-2 vaccines developed globally.

Line 68: Vaxzevria has the same component as Covishield-AstraZeneca; both are ChAdOx1-S.

Line 92-93: Was parental consent obtained for children/minor participants?

Figure 4: gray bands (95% CI) are not visible.

Reviewer 3 Report

Comments and Suggestions for Authors

This paper entitled, “Comparative Analysis of SARS-CoV-2 Antibody Responses across Global and Lesser-Studied Vaccines”, by Zambrana et. al. reports on the cross-sectional study of 3195 individuals in Nicaragua assessing the antibody response for different SARS-CoV-2 vaccines. The major findings of the study were the lower antibody titers in individuals with natural infection compared to those who had received vaccination and also the antibody responses to Soberana02 being the highest. The authors also claim that the antibody responses did not wane during the study period. Overall, this study addresses a significant gap in the literature on COVID-19 vaccine responses in diverse populations and the findings could be relevant for public health planning in these contexts. I do, however, have the following concerns:

1.      For the study sampling method, the authors should provide further information about non-probabilistic quota sampling method and the risk of bias as a result of the use of this sampling method.

2.      Further clarity on the study of the dynamics of the antibody responses is recommended. This is a cross sectional study and I assume no longitudinal sampling was carried out here. It is important to highlight this matter and discuss this further in the discussion section to avoid any potential confusion regarding the interpretation of the findings.

3.      Highlighting further research into the neutralizing effect of the antibodies induced by these vaccines could as well be very informative and should be highlighted as one of the future directions of this research.

Round 2

Reviewer 1 Report

Comments and Suggestions for Authors

Dear all,

the adjustments requested in the manuscript were made.

Best regards,

Reviewer 2 Report

Comments and Suggestions for Authors

The authors have made substantial changes in response to each of the points raised in the earlier reviews. However, there remain minor issues that should be addressed.

1)      Abstract: “Little data is available on antibody response for some SARS-CoV-2 vaccines and a lack of ability to compare vaccine responses in the same population.” This awkward sentence with a dangling phrase should be revised.

2)      It should be stated more explicitly in the Abstract that most vaccinated individuals have hybrid immunity.

3)      It would be useful to add information that Soberana is an RBD-tetanus toxoid (TT) conjugate protein vaccine, while Abdala is an RBD protein vaccine; this may be inserted on lines 68-69. The comment that “vaccination coverage against tetanus in Nicaragua is near to 100%, which may explain the incredibly high titers achieved by Soberana RBD-TT vaccine even in the first dose” would also be valuable to add in the Discussion.

Comments on the Quality of English Language

see above

Reviewer 3 Report

Comments and Suggestions for Authors

My concerns are now addressed. 

Round 3

Reviewer 2 Report

Comments and Suggestions for Authors

No additional issues noted.